# Why Do Drivers’ Collision Avoidance Maneuvers Tend to Cause SUVs to Sideslip or Rollover on Horizontal Curve and Grade Combinations?—An Analysis of the Causes Based on a Modified Multibody Dynamics Model

**DOI:** 10.3390/ijerph192315877

**Published:** 2022-11-29

**Authors:** Jinliang Xu, Wenzhen Lv, Chao Gao, Yufeng Bi, Minghao Mu, Guangxun E

**Affiliations:** 1School of Highway, Chang’an University, Xi’an 710064, China; 2Shandong Provincial Communications Planning and Design Institute Group Co., Ltd., Jinan 250101, China; 3Innovation Research Institute, Shandong Hi-Speed Group Co., Ltd., Jinan 250098, China; 4Shandong Hi-Speed Group Co., Ltd., Jinan 250098, China

**Keywords:** horizontal curve and grade combinations, multibody model, collision avoidance maneuvers, safety margin

## Abstract

The extent to which drivers’ collision avoidance maneuvers affect the safety margins of sideslip and rollover is not captured by road geometric design theory. To quantify the effects of drivers’ collision avoidance maneuvers on the safety margins of sport utility vehicles (SUVs) on horizontal curve and grade combinations, a modified 8-degree-of-freedom multibody model based on SUVs was developed. The model was then used to calculate the design safety margins of sideslip and rollover for steady states and the actual safety margins for collision avoidance maneuvers. Subsequently, the design safety margin reduction rate (the difference between the design and actual safety margins divided by the design safety margin) was calculated and used to assess the safety margins. The results showed that the safety margins of SUVs were significantly reduced by braking, lane changing, and lane changing with braking. The marginal effects indicated that the greater the deceleration and the shorter the lane change duration, the greater the effect on the safety margins, particularly the sideslip safety margin. Furthermore, when the SUV was driven at 80 km·h^−1^ on grades with a horizontal curve radius of 270 m and 400 m, the sideslip safety margin with emergency braking (deceleration over −4.5 m·s^−2^) was reduced by 71% and 21%, and the rollover safety margin was reduced by 11% and 5%, respectively. Under these conditions, an emergency lane change (lane change duration less than 2 s) caused the SUV to sideslip and reduced the rollover safety margin by 47% (curve radius 270 m) and 45% (curve radius 400 m). Therefore, drivers’ collision avoidance maneuvers are a factor that cannot be neglected in alignment design.

## 1. Introduction

Inappropriate collision avoidance behavior of drivers is the main source of risk on expressways [1,2]. To improve driving safety, scholars have conducted extensive and in-depth research in terms of geometric design consistency [3], safety speed prediction [4], and the development of active vehicle safety control systems [5]. Traffic management authorities have also introduced a series of management schemes, including speed bumps and hazard warning signs, as passive measures [6]. Despite great efforts, reports of vehicle instability incidents such as skidding and running out of lanes are still common, especially for sport utility vehicles (SUVs) with high centers of gravity [7]. This indicates that our understanding of road safety is not comprehensive enough. In fact, current road geometry design theory is based on the point–mass model, which assumes that the vehicle is driven on a curve at a constant speed and turning radius. However, the point–mass model does not consider the collision avoidance maneuvers used by the driver and the extent to which they affect the vehicle’s safety margins of sideslip and rollover. To improve driving safety, it is important to know how much drivers’ collision avoidance maneuvers affect these safety margins.

The static and dynamic response characteristics of vehicles significantly impact their stability. Khattak et al. studied the rollover strength of pickup trucks and SUVs using rollover accident data [8]. The authors concluded that the lower lateral stability (SSF) of SUVs means that they are more likely to roll over than pickup trucks. On the other hand, scholars have developed a simple point–mass model, transient BICYCLE model [9], and more complex multibody model [10] to describe the motion states and dynamics of vehicles. The lateral stability of passenger cars, SUVs, and trucks based on the SSF [11], lateral load transfer ratio [12,13], and friction margin [14], has also been studied extensively. Furthermore, it was found that the center of gravity, tire characteristics, and suspension systems of SUVs mean that their lateral stability is not as good as that of passenger cars [15].

Several studies have explored crash rates in terms of pavement conditions and alignment characteristics to some extent. Statistics show that the average crash rate is much higher on curved sections of the road than on straight sections [16] and that the average longitudinal slope is also a geometric factor that increases crash rates [17]. Similarly, poor road conditions (e.g., rain and fog) can increase road traffic crash risk. Several scholars have attempted to quantify the safety of alignment in terms of crash rates. For example, Buddhavarapu developed a horizontal curve crash severity model by integrating a database of crashes and pavement conditions [18]. Bauer used crash data from five combined horizontal and longitudinal alignments to develop a safety prediction model that could produce crash correction factors relative to horizontal curves for fatal, injurious, and property-damaging crashes [19]. Findley estimated the influence of the spatial location of horizontal curves on road safety and found that the straight-line distance between adjacent curves is a reliable predictor of crashes [20]. These results are helpful for identifying the location of road hazards and developing potential countermeasures. Unfortunately however, the research methods used to produce accident statistics cannot effectively predict the degree of impact that collision avoidance maneuvers have on road safety.

Irrational or erroneous driving behavior is the main cause of 57% of road crashes and a contributing factor to more than 90% of road collisions. Among these human factors, emergency braking and turns are the most common maneuvers and are associated with higher accident risk [1,21]. Kim et al. studied the relationship between accident rates and driving behavior and found that high accident rates are associated with drivers’ emergency braking maneuvers (braking deceleration surpassing −4.0 m·s^−2^) [2]. Khattak analyzed accident data in relation to road alignment and vehicle characteristics and found that emergency steering maneuvers increased the likelihood of vehicle rollover [1]. It can be inferred from the results of these accident investigations that drivers’ braking and steering operations significantly reduce road safety margins. To improve road safety, some recent studies have used theoretical modeling to analyze the safe speed of drivers when completing driving actions such as braking and lane changing. For example, Gallen et al. proposed an “equivalent total risk” approach where the braking distance in rain and fog is made equivalent to that in normal weather by reducing the initial speed of braking. Based on this method, they calculated the safe speed for emergency braking in rain and fog [22]. Yan developed a model for the maximum allowable speed in rain for horizontal curves during emergency braking based on the friction ellipse theory proposed by Radt [23,24]. Xin predicted the safe speed threshold to avoid rollover by simulating the dynamic rollover process of loaded trucks under sharp steering conditions [25]. Although these results can help reduce the accident risk of collision-averse driving behaviors and provide a theoretical basis for speed limit management schemes, they still cannot quantify road safety margins with collision-averse driving behaviors.

In summary, there is a gap in the literature regarding how to quantitatively assess road safety. Variation in the static and dynamic characteristics of different vehicles results in a wide variation in stability between vehicle classes. Most of the literature considers only steady state driving behavior, and studies on collision avoidance maneuvers such as braking and steering are rare. In addition, the longitudinal slope decreases the friction reserve. As such, a more accurate assessment of vehicle stability and road safety margins can be achieved by considering the combined horizontal-longitudinal alignment, but there are limited studies on this aspect. This study attempts to use the collision avoidance maneuvers of drivers to explore the safety margins of SUVs on the horizontal and vertical combinations.

To quantitatively assess the safety margins of SUVs when drivers engage in collision avoidance behaviors, an eight-degree-of-freedom (8DOF) multibody dynamics model with superelevation and longitudinal slope corrections was developed. Then, the typical collision avoidance maneuvers of drivers were discussed and safety evaluation indices were proposed. Based on this, the effects of different braking, lane changing, and combined braking and lane changing behaviors on the safety margins of SUVs were analyzed.

The rest of this paper is organized as follows: the second part develops an 8DOF multibody model with correction of superelevation and longitudinal slope variables and proposes safety margin evaluation indices under typical collision avoidance maneuvers; the third part uses the 8DOF vehicle model to analyze the safety margins of SUVs under braking, lane changing, and lane changing with braking maneuvers on horizontal curve and grade combinations and includes a discussion of marginal effects with respect to braking and lane changing; and finally, the main findings of this study and the directions for future research are summarized.

## 2. Materials and Methods

In this section, an 8DOF vehicle dynamics model is derived and corrected for superelevation and longitudinal slope. Then, the typical collision avoidance maneuvers of drivers are selected and analyzed. Finally, a safety margin assessment index and method are proposed.

### 2.1. An Improved Multibody Model with Superelevation and Longitudinal Slope Variables

The primary issue to be addressed in this research is the sideslip and rollover safety margin responses of SUVs during typical collision avoidance maneuvers on horizontal curve and gradient combinations, which falls under the field of vehicle stability. To solve this issue, a high-precision dynamic model for evaluating SUV stability is required. These models can be categorized as (improved) point–mass models, steady state (transient) BICYCLE models, and multibody models. The traditional point–mass model (the foundation of geometric design theory) and BICYCLE model can only be used to evaluate the steady state driving condition of the vehicle. Although some scholars have improved the models to some extent by considering factors such as acceleration, deceleration, slope, and road friction distribution characteristics to broaden the application scenarios, such models appear to neglect the transient response characteristics of lane change conditions. The transient BICYCLE model is very effective for vehicle lane changing and braking, but the tire model is a linear model that assumes a linear relationship between lateral force and slip angle. This assumption is applicable at small slip angles (<5°), but emergency steering and braking behavior at high speed may cause the tires to deviate from the linear characteristic. In addition, a key study from UMTRI illustrated the importance of lateral load transfer effects on the variation of lateral friction coefficients [26]. Therefore, lateral load transfer is an important factor that cannot be neglected in assessing safety margins. The more complex multibody model not only compensates for the limitations of the small slip angle assumption but can also be used to analyze the force characteristics of individual tires. Although the multibody model overcomes the limitations of the simple model, the current multibody model is mostly used for vehicle control simulation and ignores key road alignment parameters such as superelevation and longitudinal slope. Superelevation is an engineering measure implemented in the geometric design of roads to offset some of the centrifugal acceleration generated by vehicles during steering and to improve the lateral stability of the vehicle. The longitudinal slope also causes longitudinal load transfer. Therefore, it is reasonable to infer that the effects of superelevation and longitudinal slope on the safety margins of SUVs on the horizontal curve and grade combinations cannot be ignored. Given this plausibility, the multibody model was modified for superelevation and longitudinal slope and an improved multibody model with steering wheel angle, superelevation, longitudinal slope, initial speed, and braking deceleration as input variables was developed. The model was used to evaluate the safety margins of SUVs on the combinations of horizontal and vertical alignments when collision avoidance maneuvers were performed.

The improved multibody dynamics model describes the longitudinal, lateral, and yaw motions of the entire vehicle, the roll motion of the suspension system, and the rotational motion of the four wheels; therefore, it is known as the 8DOF multibody model. The assumptions made to balance modeling accuracy and computational effort are listed below. Unless otherwise specified, the International System of Units (SI) is used for each parameter of the mathematical equations in the modeling process, and radian is used as the unit to represent the angular variable.

The vehicle is symmetrical about the vertical axis;The steering wheels of the vehicle are the front wheels;It is assumed that the vehicle steers with the normal direction of all wheels pointing to the same point, i.e., the vehicle steers in accordance with Ackermann steering characteristics;The effect of tire retraction torque on tire characteristics is neglected;The yaw and pitching caused to the vehicle in addition to the longitudinal slope and superelevation are ignored.

**Vehicle dynamics model**. When the vehicle brakes or turns in the curved slope combination section, the whole vehicle will produce longitudinal, lateral, and yaw motions, and the suspension system will produce roll motion. A vehicle dynamics model taking the above motion into account has been developed in reference [27], but the superelevation and longitudinal slope of the road were not considered. In this study, these road parameters are considered. According to Figure 1a,b, the differential equation for the vehicle in the SAE coordinate system is shown in Equation (1), below:(1)$$\left\{ \begin{array}{cc} m\left({\dot{v}}_{x}-v_{y}\dot{\varphi }\right)+m_{s}h_{s}\dot{\varphi }\dot{\theta } & =\sum_{i=1}^{4}\left(F_{xi}\cos\left(\delta_{i}\right)-F_{yi}\sin\left(\delta_{i}\right)\right)-mgf-\frac{C_{d}A\rho v_{x}^{2}}{2}-mgi_{G}/\sqrt{1+i_{G}^{2}}\\ m\left({\dot{v}}_{y}+v_{x}\dot{\varphi }\right)-m_{s}h_{s}\ddot{\theta } & =\sum_{i=1}^{4}\left(F_{xi}\sin\left(\delta_{i}\right)+F_{yi}\cos\left(\delta_{i}\right)\right)+mgi_{h}/\sqrt{1+i_{h}^{2}}\\ I_{z}\ddot{\varphi }-I_{\mathrm{xz}}\ddot{\theta } & =\sum_{i=1}^{2}\left(F_{xi}\sin\left(\delta_{i}\right)+F_{yi}\cos\left(\delta_{i}\right)\right)L_{f}-\sum_{i=3}^{4}\left(F_{xi}\sin\left(\delta_{i}\right)+F_{yi}\cos\left(\delta_{i}\right)\right)L_{r}\\ & \;+\frac{T_{f}}{2}\left(-F_{x1}\cos\left(\delta_{1}\right)+F_{y1}\sin\left(\delta_{1}\right)+F_{x2}\cos\left(\delta_{2}\right)-F_{y2}\sin\left(\delta_{2}\right)\right)\\ & \;+\frac{T_{r}}{2}\left(-F_{x3}\cos\left(\delta_{3}\right)+F_{y3}\sin\left(\delta_{3}\right)+F_{x4}\cos\left(\delta_{4}\right)-F_{y4}\sin\left(\delta_{4}\right)\right) \end{array}\right.$$

According to Figure 1c, the differential equation for roll motion of the vehicle suspension system may be expressed as
(2)$$I_{x}\ddot{\theta }-I_{\mathrm{xz}}\ddot{\varphi }=m\left({\dot{v}}_{y}+v_{x}\dot{\varphi }-\frac{i_{h}g}{\sqrt{1+i_{h}^{2}}}\right)h_{\mathrm{cg}}-C_{\theta }\dot{\theta }-K_{\theta }\theta +m_{s}gh_{s}\sin\left(\theta \right)$$
where *m* is the vehicle mass and *m*_s_ is the suspension mass. *F*_x*i*_ and *F*_y*i*_ are the longitudinal and lateral tire forces, respectively. *L*_f_ and *L*_r_ are the distances from the vehicle center of gravity to the front and rear axles, respectively. *T*_f_ and *T*_r_ are the front and rear wheel spacings, respectively. *δ_i_* is the wheel turning angle. *ϕ* is the vehicle transverse sway angle. *θ* is the suspension lateral tilt angle. *h*_cg_ is the vehicle center of gravity height and *h*_s_ is the suspension system center of gravity height. *v*_x_ and *v*_y_ are the longitudinal and lateral speeds of the vehicle, respectively. *i*_G_ is the longitudinal slope, which is positive for uphill and negative for downhill. *i*_h_ is the superelevation, which is positive when the vehicle steers in the same direction as the curve, and negative when the opposite is true. *f* is the rolling resistance coefficient, and its value range is generally 0.01–0.04 [28]. *C*_d_ is the air resistance coefficient, which is obtained by the coasting test experiment. *ρ* is the standard atmospheric pressure and *g* is the acceleration of gravity. *C_θ_* is the roll damping coefficient of the suspension and *K_θ_* is the roll stiffness coefficient of suspension. *I*_z_ is the moment of inertia of the yaw of the vehicle around the z-axis, *I*_x_ is the vehicle lateral sway rotational inertia around the x-axis, and *I*_xz_ is the product of the inertia of the suspension mass around the xoz plane. *A* is the windward area. In a previous study, Wong fitted the equation for the windward area of a vehicle with an overall mass between 800 kg and 2000 kg, i.e., *A* = 1.6 + 0.00056 (*m*–765) [28]. The subscripts x, y, and z are used to distinguish the longitudinal, lateral, and normal equation parameters, respectively. The subscript *i* is used to distinguish between the left front, right front, left rear, and right rear wheels, with *i* = 1,2,3,4, respectively.

Equations (1) and (2) show that when the steering direction of the vehicle is consistent with the bending direction of the curve, the superelevation helps to reduce the cornering force of the tire and increases the anti-skid and anti-overturning stability of the vehicle. The downhill slope increases the longitudinal force of the vehicle.

**Wheel kinematics model**. According to Figure 1d, the kinematic equation for each wheel rotation can be written using the familiar moment balance:(3)$$J_{i}{\dot{\omega }}_{i}=T_{di}-T_{bi}-F_{xi}r$$
with
(4)$$T_{bi}=F_{zi}rf$$
where *T*_d*i*_ is the braking moment and *T*_b*i*_ is the rolling resistance moment. *ω_i_* is the angular speed of rotation of the wheel. *J_i_* is the moment of inertia of the wheel around the wheel center. *r* is the rolling radius of the tire.

**Tire model.** At present, many mature tire models have been developed for practice. Among them, the Pacejka magic formula is a standard method that is widely used in both vehicle dynamics research and the automobile industry to describe tire–road interactions [29]. The coefficient of the empirical formula is obtained by fitting the equation with real vehicle test data. After the coefficient fitting of the formula, the longitudinal and lateral forces during either a pure or combined slip can be estimated by inputting the tire slip rate, side slip angle, and tire normal load. In this study, the tire model is used to calculate the side friction coefficient between the tire and the road surface. The basic formula under pure slip conditions is expressed as:(5)$$y_{0}(x)=D\sin\{C\arctan[Bx-E(Bx-\arctan(Bx))]\}$$
with
(6)$$\left\{ \begin{array}{cc} Y_{0}(x) & =y_{0}(x)+S_{v}\\ x & =X+S_{h} \end{array}\right.$$

According to the input variable *X* and equation coefficients (*D*, *C*, *B*, *E*, *S*_v_ and *S*_h_), Equation (5) can reproduce the longitudinal force *F*_x_, lateral force *F*_y_, and righting moment *M*_z_ of the tire. Therefore, when the input variable *X* is the slip rate, the output variable *Y*_0_(*x*) represents *F*_x_. When the input variable *X* is a side slip angle, the output variable *Y*_0_(*x*) represents *F*_y_ or *M*_z_. *M*_z_ is not used in this paper.

When the vehicle is braking or steering in the horizontal and longitudinal combination section, the tire longitudinal force and lateral force exist at the same time, so Equation (5) must be modified to calculate the tire force in the combined slip state:(7)$$Y\left(x\right)=\left\{ \begin{array}{l} Y_{0}\left(x\right)G_{x\alpha }\left(\alpha ,\lambda ,F_{z}\right),\;\mathrm{Longitudinal}\;\mathrm{force}\\ Y_{0}\left(x\right)G_{y\lambda }\left(\alpha ,\lambda ,\gamma ,F_{z}\right)+S_{vy\lambda },\;\mathrm{Lateral}\;\mathrm{force}\; \end{array}\right.$$
where *G* (⋅) is the weighting function of the tire force in the case of combined slip. *γ* is the camber angle. *S_v_*_y_*_λ_* is the parameter of the equation. The parameters and more details of the magic formula can be found in reference [29]. The camber angle is so small that it can be ignored.

Using Equation (7), we can find values for the tire slip angle, slip rate, and normal load. The procedure for deriving these parameters can be found in the literature [30], where the slip angle of four wheels is:(8)$$\left\{ \begin{array}{l} \alpha_{1}=-\left(\delta_{1}-\arctan\left(\frac{v_{y1}}{v_{x1}}\right)\right)\\ \alpha_{2}=-\left(\delta_{2}-\arctan\left(\frac{v_{y2}}{v_{x2}}\right)\right)\\ \alpha_{3}=\arctan\left(\frac{v_{y3}}{v_{x3}}\right)\\ \alpha_{4}=\arctan\left(\frac{v_{y4}}{v_{x4}}\right) \end{array}\right.$$
where *v*_x*i*_ and *v*_y*i*_ are the longitudinal speed and lateral speed of the wheel, respectively, and are calculated by
(9)$$\left\{ \begin{array}{l} v_{x1}=v_{x}-\frac{T_{f}\dot{\varphi }}{2},\;v_{y1}=v_{y}+L_{f}\dot{\varphi }\\ v_{x2}=v_{x}+\frac{T_{f}\dot{\varphi }}{2},\;v_{y2}=v_{y}+L_{f}\dot{\varphi }\\ v_{x3}=v_{x}-\frac{T_{r}\dot{\varphi }}{2},\;v_{y3}=v_{y}-L_{r}\dot{\varphi }\\ v_{x4}=v_{x}+\frac{T_{r}\dot{\varphi }}{2},\;v_{y4}=v_{y}-L_{r}\dot{\varphi } \end{array}\right.$$

During the braking process, constraints on the friction coefficient of the tire–road interface means that the tire will slip to varying degrees in addition to rolling. The slip rate represents the proportion of the sliding component of the tire to the rolling component and is expressed as
(10)$$\lambda_{i}=\left\{ \begin{array}{l} \frac{r\omega_{i}-v_{wi}}{r\omega_{i}},\;\mathrm{if}\;v_{wi}\geq r\omega_{i}\\ \frac{r\omega_{i}-v_{wi}}{v_{wi}},\;\mathrm{if}\;v_{wi}<r\omega_{i} \end{array}\right.$$
where *λ_i_* is the slip rate of the tire. *v*_w*i*_ is the tire translation speed, which can be derived from Equation (9) and is recalled as follows:(11)$$\left\{ \begin{array}{l} v_{w1}=v_{x1}\cos\left(\delta_{1}\right)+v_{y1}\sin\left(\delta_{1}\right)=\left(v_{x}-\frac{T_{f}\dot{\varphi }}{2}\right)\cos\left(\delta_{1}\right)+\left(v_{y}+L_{f}\dot{\varphi }\right)\sin\left(\delta_{1}\right)\\ v_{w2}=v_{x2}\cos\left(\delta_{1}\right)+v_{y2}\sin\left(\delta_{1}\right)=\left(v_{x}+\frac{T_{f}\dot{\varphi }}{2}\right)\cos\left(\delta_{2}\right)+\left(v_{y}+L_{f}\dot{\varphi }\right)\sin\left(\delta_{2}\right)\\ v_{w3}=v_{x3}\cos\left(\delta_{1}\right)+v_{y3}\sin\left(\delta_{1}\right)=\left(v_{x}-\frac{T_{r}\dot{\varphi }}{2}\right)\cos\left(\delta_{3}\right)+\left(v_{y}-L_{r}\dot{\varphi }\right)\sin\left(\delta_{3}\right)\\ v_{w4}=v_{x4}\cos\left(\delta_{1}\right)+v_{y4}\sin\left(\delta_{1}\right)=\left(v_{x}+\frac{T_{r}\dot{\varphi }}{2}\right)\cos\left(\delta_{4}\right)+\left(v_{y}-L_{r}\dot{\varphi }\right)\sin\left(\delta_{4}\right) \end{array}\right.$$

The normal load transfer caused by the combination of curved slope alignment includes the load transfer between the front and rear wheels caused by the longitudinal slope and the load transfer between the left and right wheels caused by superelevation. The normal load transfer caused by braking or lane changing behavior of the driver includes the load transfer between the front and rear wheels caused by braking and the load transfer between the left and right wheels caused by steering. There is also load transfer between the front and rear wheels caused by air resistance. The dynamic normal load of each tire can be expressed as:(12)$$\{\begin{array}{l} F_{z1}=mg\frac{L_{r}}{2L}-\frac{m\left({\dot{v}}_{x}-v_{y}\dot{\varphi }+i_{G}g\right)h_{\mathrm{cg}}}{2L}-\frac{C_{d}A\rho v_{x}^{2}h_{\mathrm{cg}}}{4L}-\frac{m\left({\dot{v}}_{y}+v_{x}\dot{\varphi }-i_{h}g\right)L_{r}h_{\mathrm{cg}}}{LT_{f}}\\ F_{z2}=mg\frac{L_{r}}{2L}-\frac{m\left({\dot{v}}_{x}-v_{y}\dot{\varphi }+i_{G}g\right)h_{\mathrm{cg}}}{2L}-\frac{C_{d}A\rho v_{x}^{2}h_{\mathrm{cg}}}{4L}+\frac{m\left({\dot{v}}_{y}+v_{x}\dot{\varphi }-i_{h}g\right)L_{r}h_{\mathrm{cg}}}{LT_{f}}\\ F_{z3}=mg\frac{L_{f}}{2L}+\frac{m\left({\dot{v}}_{x}-v_{y}\dot{\varphi }+i_{G}g\right)h_{\mathrm{cg}}}{2L}+\frac{C_{d}A\rho v_{x}^{2}h_{\mathrm{cg}}}{4L}-\frac{m\left({\dot{v}}_{y}+v_{x}\dot{\varphi }-i_{h}g\right)L_{f}h_{\mathrm{cg}}}{LT_{r}}\\ F_{z4}=mg\frac{L_{f}}{2L}+\frac{m\left({\dot{v}}_{x}-v_{y}\dot{\varphi }+i_{G}g\right)h_{\mathrm{cg}}}{2L}+\frac{C_{d}A\rho v_{x}^{2}h_{\mathrm{cg}}}{4L}+\frac{m\left({\dot{v}}_{y}+v_{x}\dot{\varphi }-i_{h}g\right)L_{f}h_{\mathrm{cg}}}{LT_{r}} \end{array}$$
where *L* is the wheelbase, *L = L*_f_ + *L*_r_.

**Steering system model.** Based on the geometric analysis of Figure 1a, for a vehicle model that conforms to Ackermann’s steering characteristics, the equation for the front wheel turning angle is given by
(13)$$\left\{ \begin{array}{l} \delta_{1}=\arctan\left[\frac{2(L_{f}+L_{r})\tan\left(\delta_{f}\right)}{2(L_{f}+L_{r})-T_{r}\tan\left(\delta_{f}\right)}\right]\\ \delta_{2}=\arctan\left[\frac{2(L_{f}+L_{r})\tan\left(\delta_{f}\right)}{2(L_{f}+L_{r})+T_{r}\tan\left(\delta_{f}\right)}\right]\\ \delta_{3}=\delta_{4}=0\\ \delta_{f}=\frac{\delta_{\mathrm{sw}}}{i_{\mathrm{sw}}} \end{array}\right.$$
where *δ*_sw_ is the steering wheel corner, *δ*_f_ is the equivalent steering angle of the front wheel, and *i*_sw_ is the transmission ratio of the steering system.

**Braking system model.** The vehicle model uses the hub motor for braking. In engineering, the three-dimensional look-up table model of the motor is generally used to determine the motor output. Here, the simple motor characteristic curve is used to determine the motor output [31], which can be expressed as
(14)$$T_{di}=\frac{9550P}{n_{i}}$$
where *P* is the rated power (kW) and *n_i_* is the motor speed which is equal to *n_i_* = 30*ω_i_*/*π*.

Vehicle braking deceleration was achieved by using the braking deceleration following model, as shown in Figure 2. The basic principle is to adjust the brake pedal opening according to the error value between the actual brake deceleration *a*_real_ and the target brake deceleration *a*_x_ and the PID controller. Specifically, the hub motor adjusts the output braking torque until the error value approaches 0 such that the brake pedal aperture can ensure that the vehicle brakes with *a*_x_. In this study, only the braking case is considered, so the brake pedal opening is limited to values between −1 and 0. The adjustment results of the PID parameter values are listed in Table 1. Under the above parameters, the brake deceleration error was controlled to values with an order of magnitude of 10^−6^.

**Driving path model.** To avoid collisions with vehicles in adjacent lanes, the vehicle should not deviate from the driving lane when braking. This means that the steering wheel angle of the vehicle corresponds to the radius of the horizontal curve where the driver takes braking action. Equation (13) is an expression of the steering wheel angle which is necessary to establish the vehicle steering equation to facilitate the application of the model. The relationship between the steering wheel turning angle and the sideslip angle can be derived according to the local magnification diagram of Figure 1a [32]:(15)$$\delta_{\mathrm{sw}}=\left[\frac{L}{R_{R}}-\left(\alpha_{f}-\alpha_{r}\right)\right]i_{\mathrm{sw}}$$
with
(16)$$\alpha_{f}=\frac{\alpha_{1}+\alpha_{2}}{2},\;\alpha_{r}=\frac{\alpha_{3}+\alpha_{4}}{2}$$
where *R*_R_ is the rotation radius of the vehicle when it turns, which is a function of the superelevation and the radius of the curve: that is, $R_{R}=\sqrt{i_{h}^{2}+1}R$. *α*_f_ and *α*_r_ are the equivalent tire side slip angles of the front and rear wheels, respectively.

To capture the transient response characteristics of the vehicle when drivers change lanes or change lanes while braking, it is necessary to obtain the variation characteristics of the steering wheel angle during this process. Several steering modes in which the sine function describes the basic characteristics of vehicle steering when changing lanes have been developed [33,34,35]. As shown in Equation (17), the vehicle steering is “+” when the steering direction is consistent with the curve bending direction, and vice versa.
(17)$$\delta_{s}=\pm H\sin\left(\frac{2\pi (t-t_{0})}{\mathrm{LCD}}\right)$$

In this equation, *t* and *t*_0_ denote the time of the lane change and the start time of the lane change, respectively. *δ*_s_ is used to record the dynamic change characteristics of the steering wheel angle during the lane change.

The maximum steering range *H* of the steering wheel in Equation (17) can be obtained by the lateral offset *Y*_r_ (*Y*_r_ = 3.75 m) within the specified lane change duration (LCD):(18)$$Y_{r}=\int \left(v_{x}\sin\left(\varphi \right)+v_{y}\cos\left(\varphi \right)\right)\;dt$$

Since the object of this study is the combinations of horizontal and vertical alignments, Equation (17) is further modified by adding the offset item *δ*_sw_, which represents the steering wheel angle of the vehicle when driving in steady state on the horizontal curve and grade combinations. This modification can be expressed as
(19)$$\delta_{c}=\delta_{\mathrm{sw}}+\delta_{s}$$
where *δ*_c_ is the steering wheel angle when the vehicle changes lanes in the horizontal curve and grade combinations.

It should be added that the values *δ*_sw_ and *δ*_s_ are calculated by iteration until the resulting calculated value (*R*_real_ or *Y*_real_) is close to the target value (*R* or *Y*_r_). The specific iterative process is shown in Step 1 of Figure 3, where the iterative error is within 10^−3^ m.

### 2.2. Safety Margin Evaluation Method under Collision Avoidance Maneuvers

When drivers are faced with falling objects, water on the road, or cars that intrude into their field of vision, they will take avoidance maneuvers such as braking or changing lanes. These maneuvers increase the demand for friction coefficients, which in turn can lead to a narrowing of the friction coefficient supply–demand differential and a decrease in the safety margin. Evaluating the safety margin under collision avoidance maneuvers then becomes a Gordian knot, which is one of the objectives of this study. In this section, the typical collision avoidance maneuvers of drivers were calibrated. Then, the safety evaluation index was developed and the thresholds for non-sideslip and non-rollover of the SUV required in the evaluation index were proposed.

#### 2.2.1. *Typical Collision Avoidance Maneuvers*

**Braking.** Braking is a common collision aversion behavior. Drivers will initiate braking behavior when they feel (potential) danger. It has been found that the braking manifested by drivers is highly dependent on the degree of danger posed by the driving environment [36]. In general, the *a*_x_ taken by drivers on dry roads is approximately −1 m·s^−2^. The *a*_x_ adopted by more than 90% of drivers does not exceed −3.4 m·s^−2^. When drivers encounter unexpected obstacles, the *a*_x_ exceeds –4.5 m·s^−2^, but it is rare that the *a*_x_ exceeds −6 m·s^−2^ [37,38]. Accordingly, the following five kinds of *a*_x_ were selected for analysis:The vehicle maintains a constant speed on the curve (*a*_x_ = 0 m·s^−2^);Vehicle braking at a typical deceleration rate on the curve (*a*_x_ = −1 m·s^−2^);The vehicle enters the circular curve with the deceleration used to calculate the stopping sight distance (*a*_x_ = −3.4 m·s^−2^);Emergency braking of the vehicle on the curve (*a*_x_ = −4.5 m·s^−2^);The vehicle enters the curve at a rare emergency braking deceleration (*a*_x_ = −6 m·s^−2^).

**Changing Lanes.** In addition to lane retention braking, drivers may also make a lane change to avoid a collision. LCD is an important parameter in lane changing. Studies on LCD have found that its value range is 1.1–16.5 s, with the LCD adopted by most drivers being 5–8 s [33,39,40]. In terms of vehicle stability, a shorter LCD tends to have a smaller instantaneous turning radius and a resulting higher instantaneous lateral acceleration, which means that there are differences in driving stability under different LCDs. Therefore, LCD is also a key parameter of interest in assessing the safety margin of SUVs on horizontal curve and grade combinations. It was also found that the lane changing process is often accompanied by a slight deceleration operation with a value of approximately −1 m·s^−2^ [41]. Therefore, this study focused on examining three lane change maneuvers and two braking–lane change joint maneuvers. The LCDs for the lane change maneuvers were 6 s, 4 s and 2 s, which represent the LCD for elderly drivers, the LCD for general conditions, and the LCD for emergency conditions, respectively. The LCDs for the braking–lane change joint maneuvers were 4 s and 6 s, respectively, with *a*_x_ equal to −1 m·s^−2^.

It should be noted that the drivers’ target lane selection during a lane change determines the contribution of superelevation to vehicle stability, which can be illustrated with the help of Figure 4. If the steering direction of the vehicle is consistent with the bending direction of the curve, the superelevation becomes a stable factor which is conducive to the driving stability of the vehicle, and vice versa. Therefore, two steering conditions, marked as LC_A and LC_B, were considered.

#### 2.2.2. Safety Margin Evaluation Index

There are currently two main types of road safety evaluations. One type relies on a statistical model of predicted crash probability (the safety performance function) and uses a crash correction factor to adjust the value [20,42,43]. The many independent variables in this statistics-based approach to road safety assessment are not well targeted, and it is difficult to find specific indicators from the model that affect safety. Another category is the kinetic continuity index proposed by Lamm, which incorporates Δ*f*_R_ ≥ 0.02 for good alignment design, Δ*f*_R_ ∈ [−0.02, 0.02) for fair design, and Δ*f*_R_ < −0.02 for poor design based on the difference between the assumed value of the side friction coefficient and the demand value (Δ*f*_R_) [44]. Although this index provides a direction for safety margin evaluation, the study unfortunately did not propose a safety baseline to illustrate the degree of variation in the safety margin. Therefore, the current study proposes a design safety margin reduction rate (*f*_s_, *_k_*) to quantify the effect of drivers’ collision avoidance maneuvers on the safety margin. This reduction rate is calculated as the difference between the design safety margin and the actual safety margin divided by the design safety margin. Its mathematical form is shown in Equation (20), which covers two aspects of vehicle sideslip and rollover. *f*_s_, *_k_* = 0 means that the road safety margin is in the design state, and *f*_s_, *_k_* >0 indicates that the actual safety margin is lower than the design safety margin. When *f*_s_, *_k_* >1, the road may not provide reliable safety for the operational vehicle, and the vehicle may be unstable.
(20)$$f_{s,k}=\frac{\left(f_{t,k}-f_{b,k}\right)-\left(f_{t,k}-f_{k}\right)}{f_{t,k}-f_{b,k}},\;k=1,2$$

Equation (20) is a function of the sideslip and rollover thresholds *f*_t_, *_k_*, the baseline *f*_b_, *_k_*, and the actual demand values *f_k_*, where the subscript *k* is used to distinguish between the sideslip safety margin reduction rate and the rollover safety margin reduction rate. Here, the lateral friction coefficient, which is favored by scholars, and the lateral load transfer ratio, which can capture the instantaneous characteristics of the vehicle, were used to characterize the sideslip and rollover states of the vehicle, respectively [24,45].

From Equation (20), it can be understood that the accuracy of the calculation of *f_k_* and *f*_b_, *_k_* and the threshold of *f*_t_, *_k_* affect the accuracy of the safety margin assessment. The 8DOF multibody model guarantees that a more accurate *f_k_* (Equation (21)) is obtained because the model can record the force characteristics of each wheel in addition to capturing the dynamic change characteristics of the vehicle during the avoidance process, which is something that the simple BICYCLE and point–mass models cannot accomplish. It is very dangerous for any wheel to reach the stability limit during braking or lane changing, and the vehicle may sideslip or rollover at any time in response. Therefore, *f*_1_ takes the maximum value of the instantaneous peak of the lateral friction coefficient of each tire, while *f*_2_ is the larger value of the instantaneous peak of the lateral load transfer ratio of the front and rear wheels and takes the *f_k_* generated by a steady state driving vehicle on horizontal curves with longitudinal slope of 0 as the baseline value. The baseline value corresponds to the design value in the geometric design theory.
(21)$$f_{1}=\underset{i}{\max}\left(\max\left(\left|\frac{F_{yi}}{F_{zi}}\right|\right)\right),f_{2}=\max\left(\max\left|\frac{F_{z2}-F_{z1}}{F_{z2}+F_{z1}}\right|,\max\left|\frac{F_{z4}-F_{z3}}{F_{z4}+F_{z3}}\right|\right)$$

The determination of thresholds for the lateral friction coefficient and transverse load transfer ratio is a difficult problem during safety margin evaluation. Regarding the lateral friction coefficient, the maximum lateral friction coefficient used by AASHTO in the horizontal curve design is 0.17, which is similar to what is used in China and some European and American countries, while the United Kingdom uniformly uses 0.1 [46]. In fact, these values are based on driver comfort. By observing the vehicle speed on horizontal curves, many studies have concluded that the side friction values used in the alignment design are very conservative [47,48]. This means that even if the vehicle–road friction coefficient exceeds the maximum value, it does not mean that the situation is necessarily unsafe. The maximum tolerance of drivers for the lateral friction coefficient has been discussed in the literature and ranges from 0.22 to 0.26 [46,49]. Here, the lateral friction coefficient threshold (*f*_t_,_1_) was calibrated to 0.25. Regarding the lateral load transfer ratio, *f*_2_ = 1 indicates that one of the side tires is about to leave the ground, which is the rollover critical state. Considering the adverse effects of vehicle vibration on rollover stability, the threshold of the lateral load transfer ratio (*f*_t_,_2_) was implemented here as 0.8, which is also the value used in the literature [50]. Since this research simply provides a fresh perspective on evaluating safety margins, the precise value of the threshold has no bearing on road safety trends, and a more precise *f*_t_, *_k_* will be studied separately in the future.

In practical calculations, it is often challenging to obtain the analytical solution of the design safety margin reduction rate based on the nonlinear multibody model. Consequently, numerical solving can be considered a proven method to solve this problem. MATLAB/Simulink^®^ (Developed by MathWorks, a software developer located in Massachusetts, USA.) is commonly utilized for resolving vehicle dynamics and kinematics problems due to its capacity to significantly enhance computational efficiency [50]. Step 2 of Figure 3 shows the MATLAB/Simulink^®^ (version 9.13 for Windows) procedure for calculating the design safety margin reduction rate. The SUV parameters required to develop the 8DOF multibody dynamics model were collected from the CarSim 2020.0^®^ database and are listed in Table 2 [51]. The road models, which were obtained from the applicable requirements of the fesign Specification for Highway Alignment (JTG D20-2017) with a design speed of 80 km·h^−1^ and an assumed superelevation of 0.06, are displayed in Table 3. Since the radius of the circular curve is specified by the design speed, the initial input speed (V_d_) for the simulation is assumed to be 80 km·h^−1^, which is also the design speed. The steering wheel angle is in a stable state when the vehicle is driving steadily on a curve. If the steering wheel angle signal input during simulation is a step signal, an overshoot phenomenon which interferes with the safety evaluation results will occur; thus, the steering wheel angle (*δ*_sw_) was applied progressively from zero to a steady value during the first second. The *a*_x_ of braking behavior and the *δ*_s_ of lane change behavior were inputted during the fourth second. A study found that the braking time node during a lane change is an important factor that affects driving stability [35]; therefore, the *δ*_s_ of the braking–lane change joint behavior was inputted during the 4th second and *a*_x_ was inputted from the 4th second to the (4 + LCD)th second with a step length of 0.25 s. The simulation times for braking, lane change behavior, and braking–lane change joint behavior were six seconds, (4 + LCD) seconds and (5 + LCD) seconds, respectively. Figure 5 depicts the changes in the steering wheel turning angle and *a*_x_ during the simulation.

## 3. Results and Discussion

This research aims to evaluate the safety margins of SUVs on horizontal curve and grade combinations under drivers’ collision avoidance maneuvers. In this section, the effects of longitudinal slope and superelevation variables on safety margins were first quantitatively explored to verify the inferences made during theoretical modeling. This ws followed by an analysis and discussion of the safety margin reduction rate of the SUV on the combined horizontal–longitudinal road section under braking, lane changing, and combined braking and lane changing. To make the results easier to understand, marginal effects were used to quantify the increment of the safety margin reduction rate under braking and lane change maneuvers. The marginal effect provides the effect of a unit change in the explanatory variables on the safety margin reduction rate.

### 3.1. The Effects of Superelevation and Longitudinal Slope on the Safety Margin of the SUV

Through the theoretical analysis of Equations (1) and (2), we obtained a rudimentary grasp of how superelevation and longitudinal slope affect driving stability. To further analyze the extent to which these factors influence the safety margins of SUVs on the combined horizontal and longitudinal sections, we considered the effect of superelevation and longitudinal slope on safety margins under different driving behaviors. The results of this analysis are shown in Figure 6, where the baseline alignment conditions are *R* = 270 m, *i*_h_ = 0.06 and *i*_G_ = 0, and the baseline driving state is *a*_x_ = 0 m·s^−2^ and LCD = ∞ s (*f*_s_, *_k_* of baseline state is 0). As anticipated, for a vehicle in a steady driving state (Condition 1), neglecting the superelevation variable substantially underestimated the safety margins of sideslip and rollover by 52% and 8.2%, respectively. Due to the load transfer during braking behavior and the effect of a smaller instantaneous steering radius during lane changing behavior, *f*_s_, *_k_* or LC_A during braking should have been higher than *f*_s_, *_k_* during steady state driving. However, the *f*_s_, *_k_* values when specific collision avoidance maneuvers (Conditions 2 and 3) are performed by drivers were lower than those for Condition 1, indicating that superelevation is advantageous for collision avoidance maneuvers. Comparing LC_A (Condition 4) with LC_B (Condition 5), it was found that under the same alignment and LCD conditions, the *f*_s_, *_k_* values of LC_A were 107% and 16% lower than LC_B, respectively. This indicates that the direction of lane change is different and the effect of superelevation variables on the safety margins of the SUV is completely opposite. In a study about the critical safety speed of trucks, Xin et al. [25] found that the speed limit threshold is different when the direction of the lane change of the truck is different on the curve. In their study, when the lane change direction was consistent with the curve bending direction, the speed limit threshold was higher, that is, the allowable speed was higher when the rollover safety margin was zero. This is essentially consistent with the results of the current study. Newton-Eulerian mechanics provides a reasonable explanation for this phenomenon: namely, when the direction of lane change is the same as the direction of curve bending, the superelevation cancels out part of the lateral acceleration, but when the SUV turns in the opposite direction of curve bending, the superelevation increases the instability moment. Therefore, it is safer for the vehicle to make lane changes in the same direction as the curve. Disregarding the downhill slope leads to an overestimation of the safety margin of less than 3% compared to when it is accounted for, which is significantly less than when the superelevation variable is disregarded. In summary, ignoring superelevation variables will lead to either an overestimation (LC_B) or underestimation (steady driving, lane retention braking, LC_A) in the road safety assessment results. Thus, this research supports the claim in Section 2.1 that superelevation cannot be ignored during the theoretical modeling process. The effect of longitudinal slope on safety margins is negligible and so will not be studied separately later.

### 3.2. The Effects of Drivers’ Braking and Lane Change Maneuvers on the Safety Margin of the SUV

The *f*_s_, *_k_* for different collision aversion behaviors was computed using the safety evaluation methodology described in Section 2.2. The results of Section 3.1 demonstrated that the downhill slope had little effect on the safety margins; hence, the slope variable was disregarded in this analysis. Figure 7, Figure 8 and Figure 9 show that drivers’ collision avoidance maneuvers decrease the safety margins of SUVs on the combined horizontal and longitudinal sections. Specifically, *f*_s_, *_k_* increases with increasing *a*_x_ and decreasing LCD for the same road geometry. Moreover, greater curve radii compensate for this decrease in the safety margins.

According to Figure 7, the effect of a −1 m·s^−2^ brake deceleration on the safety margin of SUVs is limited, but the braking behavior manifested by most drivers (*a*_x_ = –3.4 m·s^−2^) significantly increases the degree of impact that this deceleration has on the safety margins. Indeed, the degree of impact increases rapidly as the radius of the curve decreases, reaching 43% for *f*_s_,_1_ and 7.1% for *f*_s_,_2_ at a radius value of 270 m. This situation is exacerbated by emergency braking (*a*_x_ = –4.5 m·s^−2^), yet *f*_s_, *_k_* <1 indicates that the drivers can still tolerate this level of braking deceleration. When *a*_x_ reaches –6 m·s^−2^, *f*_s_,_1_ is 140%. This is concerining because *f*_s_,_1_ >1 represents a dangerous condition in which the SUV is highly likely to sideslip. An analysis by Torbic et al. also shows that when SUVs enter a horizontal curve, the deceleration of −6 m·s^−2^ has a negative safety margin at many design speeds in the horizontal curve and grade combinations [52]. Furthermore, it is interesting to look at how the *f*_s_, *_k_* changed during deceleration. Table 4 shows the marginal effects of different braking manipulation levels. The values of the marginal effects show that the net effect of braking manipulation is increasing. In other words, a unit increase in deceleration increases the reduction magnitude of *f*_s_, *_k_*. It also shows that when emergency braking manipulation occurs, the possibility of sideslip or rollover increases. This result can be explained by the friction ellipse theory, which suggests that the use of braking forces reduces the available lateral friction [23].

According to Figure 8 and Figure 9, the safety margins of SUVs on the horizontal curve and grade combinations decrease to varying degrees as the LCD decreases under identical road geometry conditions. When the driver takes LC_A, the safety margin of the LCD of 4 s is guaranteed. Nevertheless, this is not always the case when LC_B is adopted. *f*_s_,_1_ exceeds one at *R* = 400 m, which is extremely detrimental to the drivers’ emergency lane change behavior. With an LCD of 2 s, the safety margin declines dramatically, and even for a horizontal curve radius of 600 m, the *f*_s_,_1_ of LC_A is significantly greater than one. Under such conditions, the coefficient of lateral friction exceeds the threshold value by a factor of 1.90, which can easily induce loss of control during tense driving.

Table 5 displays the marginal effects of varying lane change maneuver levels on the safety margins. The marginal effects demonstrate that lane change maneuvers reduce the safety margins for sideslip and rollover. The marginal effect of the lane change maneuver is unstable in terms of effect magnitude such that the marginal effect increases as the lane change duration decreases. These findings show that the safety margins during an emergency lane change maneuver are quite slim meaning that the vehicle may potentially sideslip or rollover.

Although there is no research on the safety margin response of sideslip and rollover in relation to lane change duration, there is evidence that a longer lane change duration will reduce the risk level [53], and part of the reason for this phenomenon can be found in the results of this study. The aforementioned findings also provide additional support for the conclusion based on accident statistical analysis that drivers’ emergency braking and emergency lane change behaviors are significant contributors to road crashes [1,2]. It can be concluded that emergency braking and emergency lane changes should be avoided as much as possible.

Figure 7b, Figure 8b and Figure 9b reveal that the *f*_s_,_2_ of the SUV is less than one even in the emergency avoidance condition, indicating that the SUV is not susceptible to a rollover accident with violent characteristics even if the lateral friction coefficient exceeds the driver’s tolerance limit. This is advantageous for the drivers’ safety because sideslip is frequently correctable, whereas correction of rollover requires a high level of driving skill which most nonprofessional drivers lack. The marginal effects in Table 4 and Table 5 also reflect the same result because the sideslip marginal effects and rollover marginal effects of the braking and lane change variables have a different order of magnitude. This result is supported by the findings of Yin et al., who compared the sideslip and rollover safety margins of the SUV and found that the safety factor of rollover is always higher than that of sideslip [54].

### 3.3. The Effect of Lane Change with Braking Maneuvers on the Safety Margin of the SUV

This section analyzes the effect of combined braking and lane change maneuvers on the safety margins of the SUV on the horizontal and longitudinal combined roads. Figure 10 and Figure 11 show the safety response characteristics of the SUV when drivers adopt different lane changing strategies (LCD = 4 s, 6 s) and deceleration operations (*a*_x_ = −1 m·s^−2^) at various time points. Clearly, the initiation of braking is an element that influences the safety margin. There is evidence that a vehicle’s safety margin is lowest when braking occurs 0.75 s after changing lanes [35], and our research indicates that the most dangerous braking time point during braking–lane changing behavior is related to the LCD and lane changing strategy. For LC_A, the safety margin is the lowest when the braking operation is performed at 0.75 s (LCD = 4 s) and 1.25 s (LCD = 6 s) after lane change. For LC_B, it is lowest at 2.75 s (LCD = 4 s) and 4.25 s (LCD = 6 s), and the *f*_s_, *_k_* at the most dangerous moment is more than twice that of LC_A. This contributes to Torbic’s research which we have previously discussed. The above results indicate that when the steering wheel turns in line with the direction of curve bending, the most dangerous position occurs 0.25 s before the maximum instantaneous steering angle is reached, which may be related to the hysteresis effect of the braking system (i.e., the brakes generally require between 0.2 s and 0.9 s to act) [31]. In addition, a longitudinal comparison of Figure 7, Figure 8, Figure 9, Figure 10 and Figure 11 reveal that the combined effect of braking during a lane change has a greater impact on safety margins than pure braking or lane changing. Therefore, because the required value of the side friction coefficient is lower at low speeds, it may be safer to slow down before changing lanes.

Please note that this work is only applicable to SUVs. There is evidence that the stability of trucks is significantly lower than that of SUVs [55]. This may be especially true for tractor–semitrailers that rely solely on saddle articulation and are thus more likely to skid and roll over during steering, pile up, or cause other hazardous conditions due to their large inertia, poor maneuverability, and long wheelbase [56]. Therefore, an independent investigation of the tractor–semitrailer is needed.

## 4. Conclusions

In this paper, an 8DOF multibody model modified for longitudinal slope and superelevation variables was developed to analyze the changes in the vehicle lateral friction coefficient and lateral load transfer ratio when collision avoidance maneuvers are performed by drivers. By introducing the design safety margin reduction rate index to evaluate the sideslip and rollover safety margins of the SUV on the horizontal curves and grade combinations, the study confirmed that drivers’ collision avoidance maneuvers negatively impact safety margins. The specific conclusions are listed below:Under the same road conditions, safety margins decrease sharply as braking deceleration increases. Braking with less than the designed deceleration (*a*_x_ = –3.4 m·s^−2^) has a manageable effect on the safety margins. Under emergency braking conditions (*a*_x_ = –4.5 m·s^−2^), safety margins are significantly reduced. In the special case where *a*_x_ = −6 m·s^−2^, the vehicle may sideslip on a road with a circular curve of 270 m radius, so strict speed control on curves of small radius could help prevent SUVs from sideslipping;The adverse effects of a lane change on the design safety margins are related to the duration and direction of the lane change. The marginal effects of lane changing manipulation show an increasing trend. When the lane change direction is the same as the curve bending direction, the normal lane change behavior of the driver (LCD > 4 s) has limited impact on safety, but the driver’s emergency lane change maneuver (LCD < 2 s) will cause the SUV to sideslip. Under the same lane change duration and alignment conditions, the direction of lane change determines the degree of safety margin reduction such that a lane change in the direction opposite to the curve bias results in a greater reduction in the safety margin. Therefore, emergency lane changes should be avoided as much as possible, especially when the lane change direction is opposite to the curve bending direction;With the combined effect of braking and steering, the safety margin of the SUV decreases most when braking occurs 0.25 s before the maximum instantaneous steering angle (positive value) is reached, and the degree of decrease in the design safety margin reduction rate is greater than that observed with the corresponding pure braking or lane change maneuvers;Under the same collision avoidance behavior, the safety margin of the horizontal curve and grade combinations is mainly influenced by the radius of the circular curve and superelevation, with only a small effect of the longitudinal slope on the safety margin. As the radius of the circular curve decreases, the safety margin under the same collision avoidance behavior decreases rapidly;Although the multibody simulation model is more accurate than the point–mass model, it has some limitations. For example, the currently developed models and evaluation methods cannot account for the fact that the speed of vehicles in bad weather (such as rain, fog, ice, and snow) is lower than in normal weather and that collision avoidance manipulation is also related to the driver’s personality [57]. Fortunately, the improved 8DOF multibody model and safety margin assessment method proposed in this study are flexible and allow for the modification of driving behavior parameters and the addition of environmental variables. Future field tests will collect data on real-world driving behavior, and input conditions will be modified or added to achieve a more precise assessment of sideslip and rollover safety margins;The pioneering design safety margin reduction rate index provides a new perspective for the safety evaluation of combined horizontal curve and grade combinations. The thresholds in this index do not affect the trend of safety reduction under different collision avoidance maneuvers, but rather affect the specific results of safety margin evaluations. The accurate lateral friction coefficient and lateral load transfer ratio thresholds will be studied separately in the future.

## Figures and Tables

**Figure 1 ijerph-19-15877-f001:**
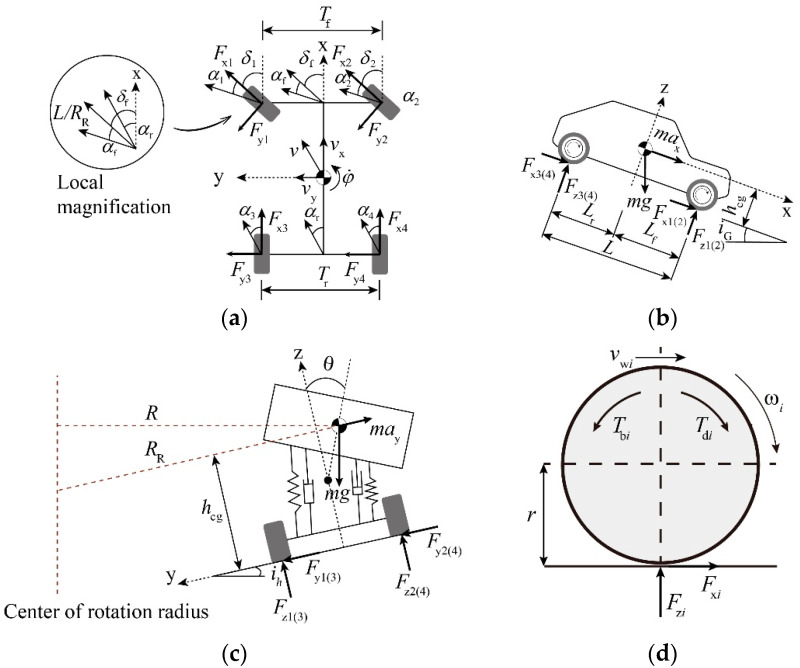
8DOF multibody model with superelevation and longitudinal slope variables. (**a**) Top view. (**b**) Side view. (**c**) Rear view. (**d**) Mechanical characteristics of the tire.

**Figure 2 ijerph-19-15877-f002:**
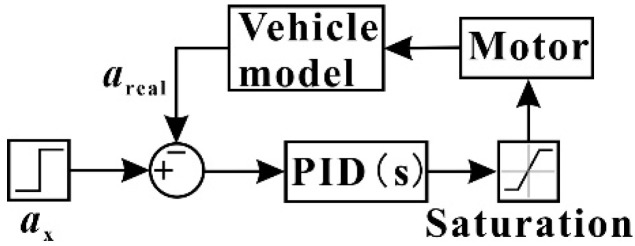
Brake deceleration following model.

**Figure 3 ijerph-19-15877-f003:**
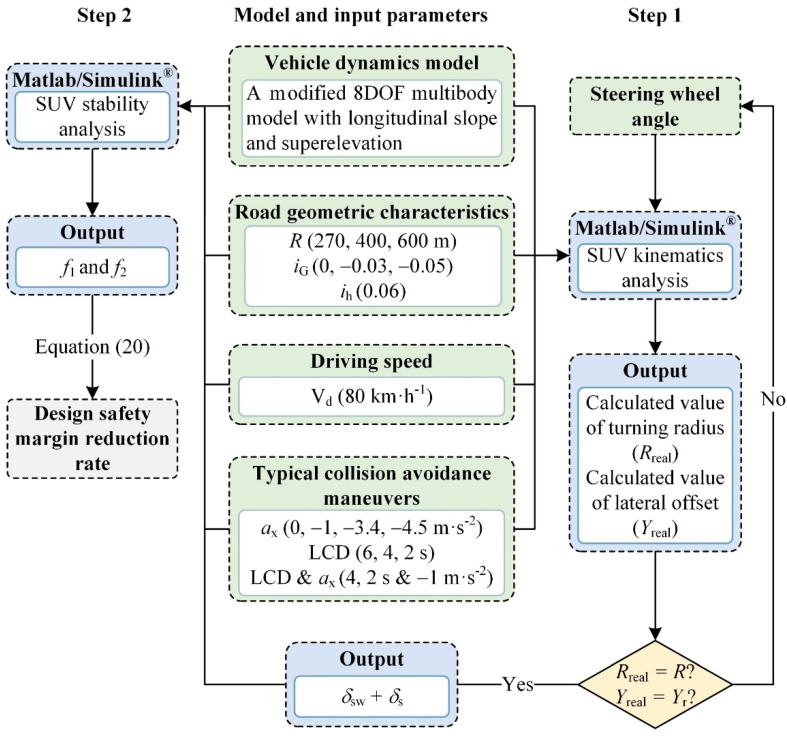
Calculation process for the steering wheel angle and design safety margin reduction rate.

**Figure 4 ijerph-19-15877-f004:**
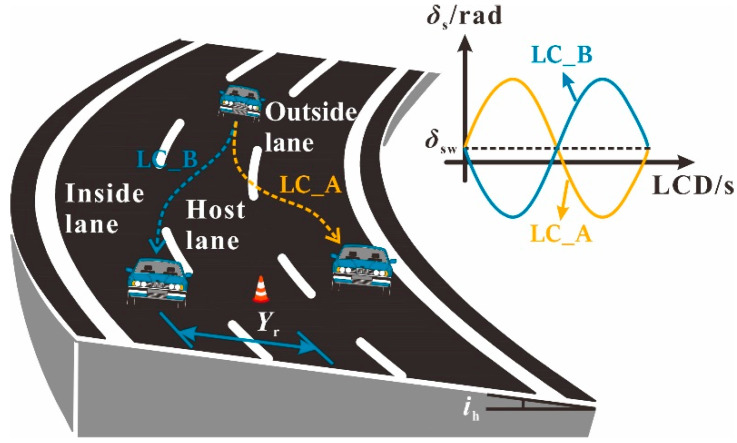
Lane changing process.

**Figure 5 ijerph-19-15877-f005:**
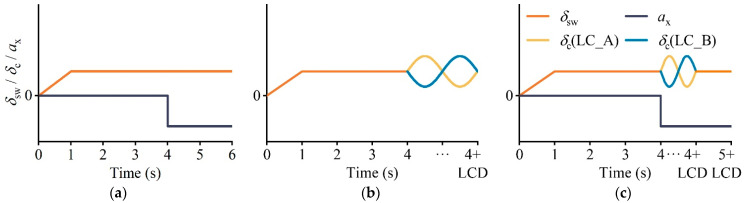
Input time point of the steering wheel angle and deceleration. (**a**) Pure braking manipulation. (**b**) Pure lane change manipulation. (**c**) Lane change with braking manipulation.

**Figure 6 ijerph-19-15877-f006:**
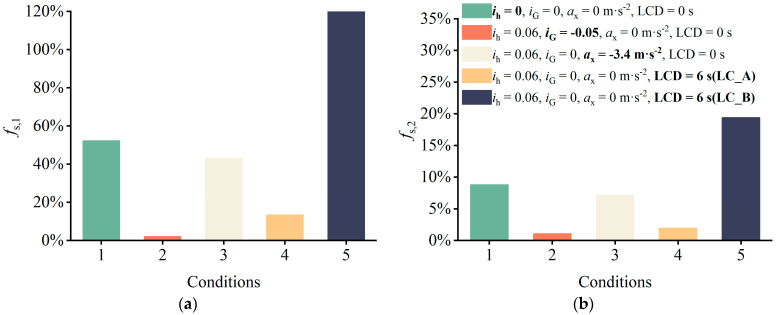
Effects of superelevation and longitudinal slope on the safety margin of the SUV. (**a**) Sideslip margin reduction rate. (**b**) Rollover margin reduction rate.

**Figure 7 ijerph-19-15877-f007:**
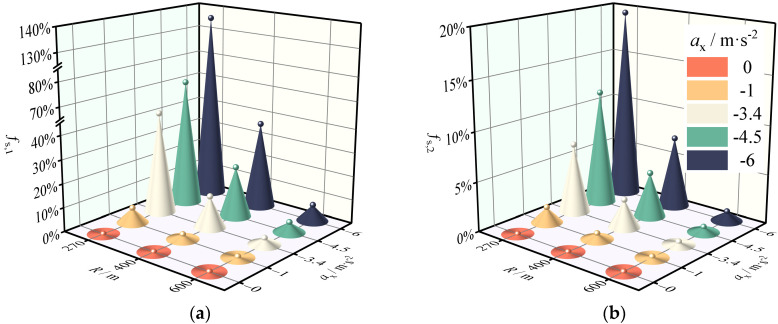
Safety margin reduction rate of the SUV on the horizontal curve and grade combinations under braking. (**a**) Sideslip margin reduction rate. (**b**) Rollover margin reduction rate.

**Figure 8 ijerph-19-15877-f008:**
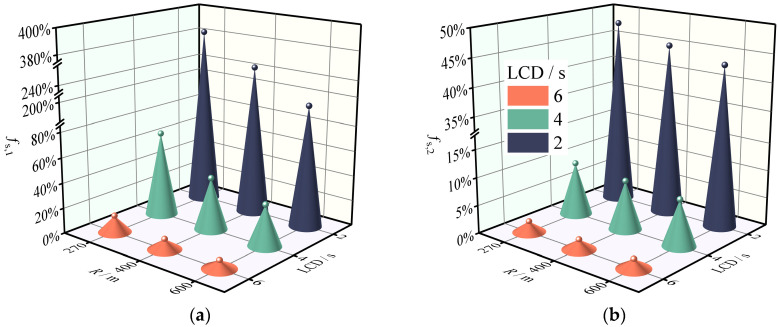
Safety margin reduction rate of the SUV on the horizontal curve and grade combinations under LC_A. (**a**) Sideslip margin reduction rate. (**b**) Rollover margin reduction rate.

**Figure 9 ijerph-19-15877-f009:**
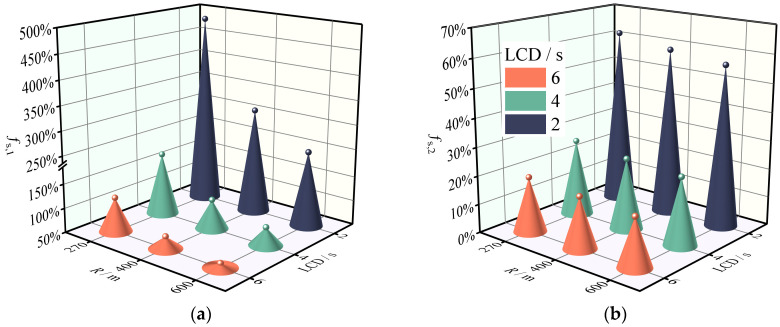
Safety margin reduction rate of the SUV on the horizontal curve and grade combinations under LC_B. (**a**) Sideslip margin reduction rate. (**b**) Rollover margin reduction rate.

**Figure 10 ijerph-19-15877-f010:**
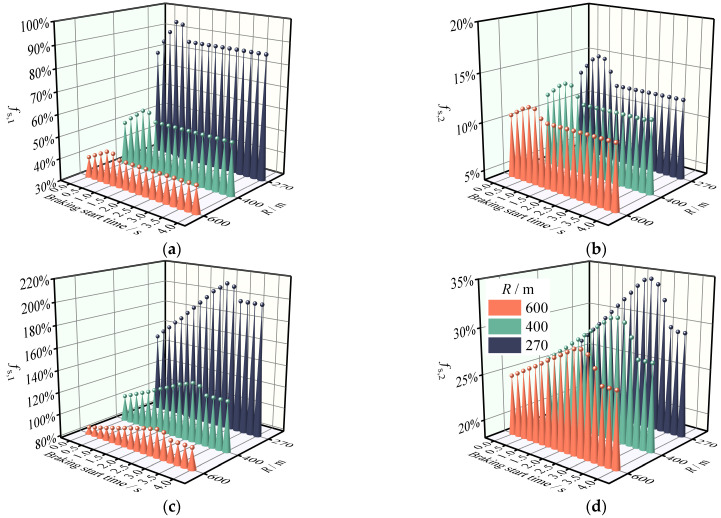
The SUV’s safety margin reduction rate on horizontal curves and grade combinations when braking and lane changing are combined (LCD = 4 s). (**a**) Sideslip margin reduction rate for LC_A with braking maneuver. (**b**) Rollover margin reduction rate for LC_A with braking maneuver. (**c**) Sideslip margin reduction rate for LC_B with braking maneuver. (**d**) Rollover margin reduction rate for LC_B with braking maneuver.

**Figure 11 ijerph-19-15877-f011:**
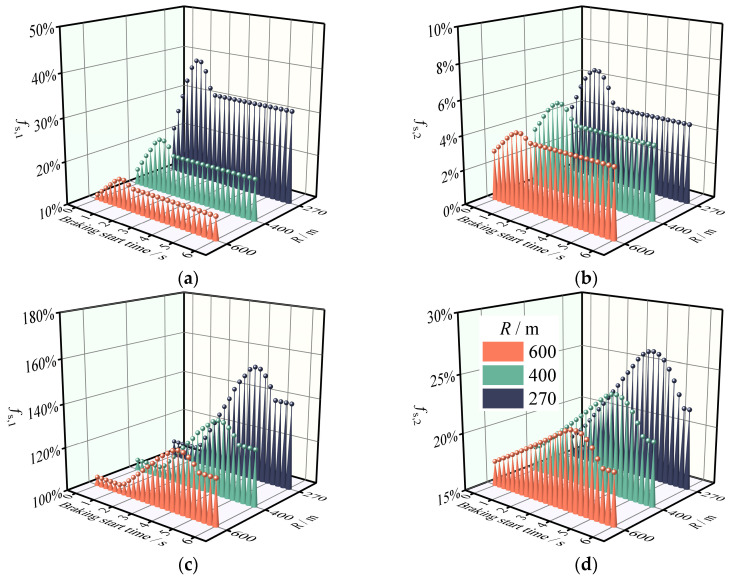
The SUV’s safety margin reduction rate on horizontal curves and grade combinations when braking and lane changing are combined (LCD = 6 s). (**a**) Sideslip margin reduction rate for LC_A with braking maneuver. (**b**) Rollover margin reduction rate for LC_A with braking maneuver. (**c**) Sideslip margin reduction rate for LC_B with braking maneuver. (**d**) Rollover margin reduction rate for LC_B with braking maneuver.

**Table 1 ijerph-19-15877-t001:** PID parameter optimization results *.

P	I	D
0.3	25	0.0001

* The P, I and D in the table denote the proportional, integral and differential parameters of the PID controller, respectively.

**Table 2 ijerph-19-15877-t002:** **Description of** SUV parameters.

Symbol/Unit	Parameters	Symbol/Unit	Parameters
*m*/kg	1530	*g*/(m·s^−2^)	9.81
*m*_b_/kg	1430	*I*_z_/(kg·m^2^)	2059.2
*m*_w_/kg	25	*I*_x_/(kg·m^2^)	700.7
*T*_f_/m	1.565	*I*_xz_/(kg·m^2^)	0
*T*_r_/m	1.565	*I*_tire_/(kg·m^2^)	1.9
*L*_f_/m	1.05	*K_θ_* /(N·m·rad^−1^)	145,330
*L*_r_/m	1.61	*C_θ_* /(N·m·rad^−1^)	4500
*h*_cg_/m	0.65	*P*/kW	160
*h*_s_/m	0.65	*T*/(N·m)	1000
*r*/m	0.357	*i*_sw_/(nondimensional quantity)	20

**Table 3 ijerph-19-15877-t003:** Road geometric parameters.

Number	*R*/m	*i* _G_	Number	*R*/m	*i* _G_	Number	*R*/m	*i* _G_
1	270	0	4	400	0	7	600	0
2	270	−0.03	5	400	–0.03	8	600	–0.03
3	270	–0.05	6	400	–0.05	9	600	−0.05

**Table 4 ijerph-19-15877-t004:** Marginal effect of braking manipulation *.

*R*/m	*a*_x_∈[0, −1]	*a*_x_∈[−1, −3.4]	*a*_x_∈[−3.4, −4.5]	*a*_x_∈[−4.5, −6]
270	−0.0621 (−0.0157)	−0.1535 (−0.0232)	−0.2623 (−0.0422)	−0.4261 (−0.0517)
400	−0.0203 (−0.0059)	−0.0466 (−0.0094)	−0.0748 (−0.0156)	−0.1009 (−0.0182)
600	−0.0040 (−0.0009)	−0.0096 (−0.0008)	−0.0145 (−0.0026)	−0.0171 (−0.0028)

* The sideslip marginal effects of braking manipulation are shown by the values outside the parentheses, while the rollover marginal effects are shown by the values inside the parentheses.

**Table 5 ijerph-19-15877-t005:** Marginal effect of lane changing manipulation *.

*R*/m	LC_A		LC_B	
	LCD∈[6, 4]	LCD ∈[4, 2]	LCD∈[6, 4]	LCD ∈[4, 2]
270	−0.2698 (−0.0382)	−1.5783 (−0.1911)	−0.2851 (−0.0373)	−1.5306 (−0.1760)
400	−0.1673 (−0.0356)	−1.0099 (−0.1811)	−0.1779 (−0.0349)	−0.9954 (−0.1692)
600	−0.1315 (−0.0331)	−0.8066 (−0.1749)	−0.1406 (−0.0334)	−0.8012 (−0.1648)

* The sideslip marginal effects of lane changing manipulation are shown by the values outside the parentheses, while the rollover marginal effects are shown by the values inside the parentheses.

## Data Availability

The complete data are available within the article.

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
