# Peer review of "Why Do Drivers’ Collision Avoidance Maneuvers Tend to Cause SUVs to Sideslip or Rollover on Horizontal Curve and Grade Combinations?—An Analysis of the Causes Based on a Modified Multibody Dynamics Model"

_ijerph, 2022, doi:10.3390/ijerph192315877_

Round 1

Reviewer 1 Report

1)P3  line129 “This study.....performed.” , the paragraph needs edited, the logistic should be displayed more clearly.

2) The 8DOF multibody model should be defined more clearly to understand.

3) The data in this study is obtained from Matlab simulation, there are much differential between simulation and real road, because it can not consider the drivers style and driving environments.

4) The discussion section should be compared with the previous studies.

5) Did the study calculate the marginal effect? 

Reviewer 2 Report

The authors study effects of drivers maneuver on the safety of the vehicle. To measure the effects of drivers' collision avoidance maneuver on the safety margin of sport utility vehicles on horizontal curve and grade combinations, an 8-degree-of-freedom multibody model is proposed.  This model is used to quantify the design safety margin of sideslip and rollover for steady-states and the actual safety margin for collision avoidance maneuvers.

The paper is well written, presents interesting results and fits the scope of the journal. English is good. I recommend publishing the paper after considering the following minor comments:

1.       Please state clearly if equations (1), (2) were obtained by the authors. If not, please provide corresponding references. Similarly, the other expressions in Section 2.

2.       In Section 4 Conclusions the authors analyze advantages of the proposed approach. However, critical analyses of the limitations of their modeling and simulation scheme would improve the quality of the paper.  
